# Current Molecular Combination Therapies Used for the Treatment of Breast Cancer

**DOI:** 10.3390/ijms231911046

**Published:** 2022-09-20

**Authors:** Yiling Wang, Audrey Minden

**Affiliations:** Susan Lehman Cullman Laboratory for Cancer Research, Department of Chemical Biology, Ernest Mario School of Pharmacy, Rutgers, The State University of New Jersey, Piscataway, NJ 08854, USA

**Keywords:** breast cancer, cellular pathways, molecular drugs, combination therapy

## Abstract

Breast cancer is the second leading cause of death for women worldwide. While monotherapy (single agent) treatments have been used for many years, they are not always effective, and many patients relapse after initial treatment. Moreover, in some patients the response to therapy becomes weaker, or resistance to monotherapy develops over time. This is especially problematic for metastatic breast cancer or triple-negative breast cancer. Recently, combination therapies (in which two or more drugs are used to target two or more pathways) have emerged as promising new treatment options. Combination therapies are often more effective than monotherapies and demonstrate lower levels of toxicity during long-term treatment. In this review, we provide a comprehensive overview of current combination therapies, including molecular-targeted therapy, hormone therapy, immunotherapy, and chemotherapy. We also describe the molecular basis of breast cancer and the various treatment options for different breast cancer subtypes. While combination therapies are promising, we also discuss some of the challenges. Despite these challenges, the use of innovative combination therapy holds great promise compared with traditional monotherapies. In addition, the use of multidisciplinary technologies (such as nanotechnology and computer technology) has the potential to optimize combination therapies even further.

## 1. Classification of Different Subtypes of Breast Cancer

Breast cancer is not a single disease, but rather it can be categorized into different subtypes [1,2]. The main subtypes are (1) luminal A, (2) luminal B, (3) HER2 positive, (4) basal-like, and (5) normal-like [3,4,5,6]. These subtypes can all be divided even further. Luminal A and Luminal B subtypes usually have high levels of estrogen receptor (ER) and/or progesterone receptor (PR), and are thus considered to be hormone receptor positive. Luminal B is usually also positive for Her2. Luminal A is usually low in Ki67 (a marker for cell proliferation), whereas luminal B often has high Ki67 [7,8]. Normal-like breast cancer usually shares characteristics with luminal A breast cancer, with some slight differences in overall genetic makeup [8,9,10]. Normal-like and luminal A tumors have similar classical immunohistochemistry markers, such as ER+, PR+, HER2−, and low KI67 expression [11]. Other than this, the overall gene expression pattern of normal-like breast cancer is similar to that of normal breast tissue. Due to their similarities, these two types of cancer are difficult to distinguish, although a recent study found that they may differ in their expression signatures of lncRNAs (long non-coding RNAs) [12]. Furthermore, normal-like breast cancer generally has a better prognosis than luminal A [8,11,13]. Basal-like breast cancers are often triple-negative (TNBC), in that they lack expression of ER, PR, and Her2 [1,3,4,5,6,7,8,9,14,15]. Within this group, TNBC itself can be further divided into four subtypes: basal-like 1, basal-like 2, mesenchymal, and luminal androgen receptor, but 80% of TNBC breast cancers overlap with the basal-like subtype [7,9,16]. Among the different types of breast cancer, those that overexpress ER (ER-positive) are the most common, and this type of cancer often comes with diverse gene mutations, which require multiple treatments [10]. Overexpression of HER2 appears in 12–20% of breast cancers, where the HER2 gene is frequently amplified [14]. Among all the different subtypes, TNBC is particularly difficult to treat because of its lack of recognizable markers and hence the lack of specific druggable targets. Overall, the heterogeneous nature of breast cancer makes the development of treatments quite complex.

## 2. Introduction to Combination Therapies for the Treatment of Breast Cancer

Breast cancer is the most common cancer in the world [1]. Even though significant advances in research have been made, breast cancer has high incidence and mortality rates. Currently, various therapies are used to treat breast cancer, including endocrine therapies, cytotoxic chemotherapies, and targeted therapies. A schematic illustration of the different types of breast cancer treatment is shown in Figure 1. While a wide range of treatments have been developed, the overall survival rate remains low, and some types of breast cancer, such as triple-negative breast cancer, are particularly difficult to treat [17]. Despite monotherapies (such as targeted therapy, hormone therapy, immunotherapy, or chemotherapy) having a good efficiency for treating some breast cancer patients, their effectiveness sometimes weakens over time, and some patients develop resistance to treatment. While there has been some success with the use of monotherapies, some patients do not respond well, especially those with metastatic breast cancer [17]. Approximately 75% of breast cancer deaths are caused by metastases, and this has remained consistent over recent years [18]. Even among patients who initially respond to monotherapy treatment, many eventually relapse [19,20,21]. Development of more effective and targeted treatments for breast cancer is thus urgently needed.

A new and promising idea is the use of combination therapy, where two or more agents are used to treat the disease. Various types of combination therapy are currently being used in the clinic and in clinical trials. Several new combinational approaches have been developed. As this type of treatment becomes more widespread, technological advances are likely to lead to even more innovative approaches. Furthermore, supportive data such as nanotechnology, DNA sequencing technologies, and computational analysis can be used to optimize different combination approaches. Combination therapies can have significant advantages over monotherapies, as discussed in more detail later in this review. Importantly, however, challenges remain, including preventing drug resistance, toxicity, drug–drug interaction risks, and insufficient overall survival analysis. Furthermore, there remains a need for combination therapies to be optimized based on the responses of different patients. It is therefore important to understand the different combination approaches that are currently used either in the clinic or in clinical trials, in order to assess the best approaches for treating patients. Here we focus on discussing current combination therapies, with the ultimate goal of improving outcomes for breast cancer patients. Notably, our review focuses only on combination therapies using two or more completely different types of drugs which target different pathways. Most of the combination therapies we discuss are currently used in clinical trials or preclinical studies [3,22,23,24,25], while some have been approved by the FDA. Some of the important combination therapies are summarized in Table 1 [26]. These types of therapies are the main topic of this review. Before discussing the combination therapies in detail, we first summarize some of the current monotherapies that are in use, and subsequently discuss the potential advantages that combination therapies may have over these more traditional treatments.

## 3. Current Molecular Drugs for Monotherapy (Single-Agent)

Drugs that are directed against specific targets have generally been shown to be more effective and to have less toxicity compared with standard therapy such as chemotherapy and radiation therapy [3]. Some of these drug treatments are described below.

### 3.1. Hormone Therapy/Endocrine Therapy

In the USA, more than 70% of breast cancer patients exhibit high expression of estrogen receptor-alpha (*ER*α) [46]. Hormone therapy, also called endocrine therapy, improves the survival of patients with hormone receptor (HR)-positive breast cancer. Endocrine therapy includes the use of antiestrogens and aromatase inhibitors. Antiestrogens generally act by modulating estrogen receptors. The major types of antiestrogens are the “selective estrogen receptor modulators” (SERMs) and “selective estrogen receptor downregulators” (SERDs) [37,47]. Currently, SERMs and SERDs (e.g., Tamoxifen, Toremifene, Raloxifene, Fulvestrant) are both widely used for the treatment of breast cancer, particularly among postmenopausal women [48]. Aromatase inhibitors (AIs) comprise a separate category of inhibitors designed to block estrogen. AIs inhibit the conversion of androgens to estrogen in peripheral tissues [37,49] and can also be used to treat ER-positive breast cancer [1,50,51].

Some hormone receptor positive breast cancers are also positive for the hormone progesterone. Since many ER-positive breast cancers are also PR-positive, treatments that block ER are often used to treat ER+/PR+ breast cancer. In a minority of cases, tumors can be PR-positive but ER-negative [1,17,49,52,53,54]. 

Although hormone therapy has been shown to elicit beneficial effects in many breast cancer patients, it may also be harmful to certain tissues (eg. bone, heart, and brain) due to long term estrogen stimulation [47]. Furthermore, some patients can have antagonist reactions to hormone treatment [47,48]. Thus, side effects can limit the usefulness of hormone therapy, and the development of newer combination therapies is an ongoing effort to solve these problems.

### 3.2. Targeted Therapies

Several molecular targeted therapies are being used or explored to target various subtypes of breast cancer. Such therapies include PI3K/AKT/mTOR inhibitors and AMPK activators (which inhibit the mTOR pathway), such as Metformin and Demethoxycurcumin) [2,55,56,57], PARP inhibitors (Olaparib, Talazoparib, Veliparib, Iniparib), mTOR inhibitors (Everolimus), pan-PI3K inhibitors (e.g., Buparlisib, Alpelisib, Pictilisib, Taselisib), CDK4/6 inhibitors (Palbociclib, Ribociclib, Abemaciclib), and other tumor-specific molecular target drugs (e.g., Deubiquitinases), and microRNAs (e.g., miRNA-200c, miRNA-21, miR-185-5p) [3,58,59,60]. These agents have been shown to significantly increase overall survival in patients who are resistant to endocrine therapy, and to be more effective than chemotherapy. For example, patients with mutations in the breast cancer susceptibility genes 1 and 2 (BRCA1 and BRCA2) and HER2-negative metastatic breast cancer were treated with the monotherapy drug Olaparib (oral PARP inhibitor) [61]. The median progression-free survival (PFS) of the patients treated with Olaparib was 2.8 months longer than those treated with chemotherapy. The risk of disease progression or death was 42% lower than with chemotherapy.

### 3.3. Immunotherapies

Several types of immunotherapies are currently used to treat breast cancer, especially TNBC. TNBC accounts for 15–20% of all breast cancers [1]. Because TNBC lacks expression of ER, PR, and Her2, TNBC patients do not benefit from endocrine therapy or HER2-targeted therapy. Unlike other therapies, immunotherapy is a strategy for destroying abnormal cells through the patient’s own immune system, leading to suppression of tumor growth [1,62]. Immunotherapies include immune checkpoint inhibitors, T-cell transfer therapy, monoclonal antibodies, treatment vaccines, and immune system modulators. Immune checkpoint inhibitors have been used in trials to block checkpoint proteins from binding with their partner proteins, resulting in activation of the T-cell response and subsequent killing of cancer cells. Commonly used immune checkpoint inhibitors include the antagonists of the programmed cell death-1/programmed death ligand-1 (PD-1/PD-L1) such as Avelumab, Atezolizumab, Tezolizumab, and Pembrolizumab, and those of cytotoxic T-lymphocyte-associated protein 4 (CTLA-4), such as Tremelimumab and Ipilimumab. PD-1, PD-L1, and CTLA-4 inhibitors are often used to treat metastatic TNBC, HR-positive (HR+), and HER2-negative (HER2−) breast cancers, respectively [1,46,63].

One area where immunotherapy has been particularly prominent is in the treatment of Her2+ breast cancer. Of particular importance is the use of monoclonal antibodies against HER2 such as Trastuzumab (Herceptin), Trastuzumab-DM1 (T-DM1), and Pertuzumab [41]. Currently, Trastuzumab is approved by the FDA for use in the first-line treatment of HER2+ metastatic breast cancer, either as monotherapy or in combination with chemotherapy [64]. Monoclonal antibodies such as Trastuzumab have shown favorable results, by promoting an immune response (activation of T cells, natural killer cells, and macrophages) and producing cytotoxic antitumor cell effects in patients [65,66]. Moreover, Trastuzumab downregulates the HER2, thus reducing Her2+ cancer cell growth [40,65,67,68,69]. T-DM1 is a combination of Trastuzumab and Emtansine (DM1, a cytotoxic agent). Compared with Trastuzumab, T-DM1 significantly improved invasive disease-free survival in patients with HER2+ early breast cancer; distant recurrence in the T-DM1 group was 10.5%, compared to 15.9% in the trastuzumab group [70]. T-DM1 is thus more effective than Trastuzumab alone at reducing recurrence in invasive Her2+ breast cancer [70,71]. Although T-DM1 is in phase-3 clinical trials in the USA, it has already been approved in the European Union for patients with HER2-positive unresectable locally advanced or metastatic breast cancer [72].

## 4. Combination Therapies and Breast Cancer Treatment

While some positive results have been obtained using monotherapies, as described above, monotherapies are not always effective. For this reason, combination therapies are being developed which combine two or more therapeutic agents. The goal is to improve patient outcomes by targeting multiple pathways, and to reduce side effects and development of resistance. The remainder of this review addresses the various types of combination therapies available or under development. 

### 4.1. Combination of Molecular Targeted Therapy and Endocrine Therapy

Although endocrine therapy is effective, some patients develop resistance. The use of molecular targeted therapy alongside endocrine therapy is therefore being explored. Here, we discuss endocrine therapy combined with molecular targeting compounds. These include compounds that target mTOR, PI3K, CDK4/6, and dual PI3K/mTOR inhibitors. We discuss this combination approach for the treatment of breast cancer, especially metastatic HR+, HER2− advanced-stage breast cancer.

#### 4.1.1. mTOR Inhibitor plus Hormone Receptor Inhibitors

Targeted therapy combined with endocrine therapy was assessed in patients with HR+, advanced-stage breast cancer who built up resistance to endocrine therapies [50]. Letrozole (an AI) alone was compared with Letrozole combined with Everolimus (a selective mTOR inhibitor). In a phase-2 clinical trial, the PFS increased from 9.0 months to 22.0 months when the combination treatment was used, compared with Letrozole alone [50,73,74]. The results from this trial indicated that combined inhibition of the mTOR signaling pathway and the estrogen receptor might suppress or delay resistance to endocrine therapy in patients with advanced breast cancer [50]. In similar studies, Everolimus was combined with endocrine therapy in postmenopausal women with endocrine-resistant HR+, HER2− breast cancer. This resulted in significant improvement in PFS and objective response rate, compared with endocrine therapy alone [75].

#### 4.1.2. CDK 4/6 Inhibitors plus Hormone Receptor Inhibitors

Combing endocrine therapy (such as aromatase inhibitors [Letrozole or Anastrazole], or Fulvestrant) with CDK 4/6 inhibitors (such as Palbociclib or Abemaciclib) was beneficial for increasing endocrine sensitivity in patients with HR+, HER2− advanced or metastatic breast cancer [76]. The median PFS was 28 months for patients in the aromatase inhibitor and CDK 4/6 inhibitor combination therapy group, compared with 14.9 months for the group receiving placebo plus endocrine therapy. The median PFS in the Fulvestrant and CDK 4/6 inhibitor combination therapy group was 22.4 months, higher than endocrine therapy alone, and this is still likely to have been higher, due to patients who were surviving at the time the report was made [76].

Results similar to those described above were also observed in several other clinical trials [77,78,79]. For example, adjuvant CDK4/6 inhibitors (Ribociclib [80], Abemaciclib, or Palbociclib [78]) in combination with endocrine therapy, were more effective than endocrine therapy alone in patients with HR+,HER2- early breast cancer. However, not all combination trials have been as successful. For example, in a separate study, adding Palbociclib to endocrine therapy (either anti-estrogen or AI) for one year did not improve disease-free survival in patients with residual invasive disease after neoadjuvant chemotherapy [79]. This trial was initially carried out for one year but remains ongoing with the hope that longer term treatment will improve the outcome. 

#### 4.1.3. PI3K Inhibitors or Dual PI3K/mTOR Inhibitors plus Aromatase Inhibitor

Activation of PI3K due to lack of PTEN protein can trigger resistance to endocrine therapy in ER-negative breast cancer. PTEN is a tumor suppressor, which can lead to resistance to endocrine therapy through regulation of the PI3K pathway [81,82]. Repression of the PI3K signaling pathway is therefore considered to be a promising approach for treatment of breast cancers that are resistant to endocrine therapy. The PI3K inhibitor (Pilaralisib) and a dual PI3K/ mTOR inhibitor (Voxtalisib) in combination with Letrozole (an AI) were each shown in clinical trials to be effective in patients with HR+, HER2− nonsteroidal-AI-refractory recurrent or metastatic breast cancer [83]. In that study, 33% of patients in the Pilaralisib + Letrozole group, and 22% of patients in Voxtalisib + Letrozole group had a PFS of 24 weeks. Although efficacy was only modestly improved, Pilaralisib was more effective than Voxtalisib. This may be due to a greater pharmacodynamic impact of Pilaralisib on glucose metabolism, compared with that of Voxtalisib. Both PI3K and mTOR play important roles in glucose metabolism as well as other cellular processes. One explanation for the above results is that the impairment of glucose metabolism by PI3K/mTOR inhibitors might benefit the anti-tumor activity in drug combinations [84]. This study showed that PI3K inhibitors in combination with AI might have significant value in endocrine therapy resistance. However, not all results from endocrine therapy combined with PI3K inhibitors or dual PI3K/mTOR inhibitors have been promising, and challenges remain for the development of this type of combination treatment. In a phase-2 trial, for example, the combination of Pictilisib (a PI3-kinase inhibitor) and Fulvestrant did not appear to improve results for patients with AI-resistant advanced HR+ breast cancer [85]. However, this could be because Pictilisib can have significant toxicity, so approximately 45% of patients needed a reduced dose or had to discontinue treatment.

#### 4.1.4. PI3K/AKT/mTOR Inhibitors plus Hormone Receptor Inhibitors

Activation of the PI3K/AKT/mTOR signaling pathway commonly occurs in breast cancer [81]. PI3K/AKT/mTOR inhibitors are the second-line treatment in postmenopausal women with HR+, HER2− metastatic breast cancer [81]. The uses of three CDK4/6 inhibitors and PI3K/AKT/mTOR inhibitors along with hormone therapy (Fulvestrant) were explored. The CDK4/6 inhibitors were Palbociclib, Ribociclib, and Abemaciclib, while the PI3K/AKT/mTOR inhibitors included Everolimus (mTOR inhibitor), Capivasertib (AKT inhibitor), and Pictilisib or Buparlisib (PI3K inhibitors). The probability of a treatment’s efficacy and safety was analyzed by the “surface under the cumulative ranking curve” (SUCRA). The SUCRA value from that particular study indicated that Abemaciclib plus Fulvestrant was the best regimen (85.29%), Capivasertib plus Fulvestrant had comparable efficacy (80.89%), followed by Ribociclib plus Fulvestrant (75.83%). The lowest SUCRA value was for Alpelisib plus Fulvestrant (6.797%), for patients with PIK3CA-mutated HR+, HER2− advanced breast cancer [86,87]. 

### 4.2. Combination of Immunotherapy and Chemotherapy

Although immunotherapy has been used to treat various cancers, it is less widely used than surgery and traditional therapies (chemotherapy or radiation therapy). While immunotherapy can be successful, not all patients respond to it. Furthermore, some patients treated with a combination of immunotherapy drugs may develop immune-related diseases, such as fulminant myocarditis [88].

Recent research has explored combinations of immune checkpoint inhibitors with targeted therapy, chemotherapy, or radiation therapy to overcome some of the problems that can occur with immunotherapy alone. The US FDA has approved a number of immune checkpoint inhibitors for breast cancer treatment, such as the CTLA-4 inhibitor Ipilimumab, PD-L1 inhibitors, as well as PD-1 inhibitors (which include Atezolizumab, Durvalumab, and Avelumab). Several results from clinical trials indicate that Atezolizumab (PD-L1 antibody) [89,90,91] and Pembrolizumab (PD-1 antibody) [62,92,93,94] are both promising immunotherapies when combined with chemotherapy in first-line treatment for metastatic and advanced TNBC [90]. 

The combination of the monoclonal antibody Atezolizumab with the chemotherapeutic drug nab-paclitaxel (A + nP) was shown to be beneficial for some patients with metastatic triple negative breast cancer. The tumor microenvironment and biomarkers can play an important role in patient response. Various patients were treated with the combination (A + nP) or with nP + placebo (P + nP) to determine which patient groups are most likely to benefit from this type of combination treatment. PFS and OS were assessed in relation to tumor microenvironment and specific biomarkers [62]. These included PD-L1 expression in immune cells (PDL1-IC) or tumor cells, intratumoral CD8, stromal tumor-infiltrating lymphocytes, and BRCA1/2 mutation which is commonly found in breast cancer. The results from the study indicated that patients who benefited most from the combination treatment were those with a rich tumor immune microenvironment, but this was restricted to patients whose tumors were PD-L1 IC+ (IC+: PD-L1 > 1%). This illustrates the importance of evaluating various biomarkers and the tumor microenvironment before selecting specific treatments. 

### 4.3. Combination of Immunotherapy and Endocrine Therapy

Endocrine therapy is often used to treat HR-positive breast cancer patients. Interestingly, in addition to its effects on hormone levels, endocrine therapy can also have an immunomodulatory effect by regulating the infiltration of immune cells into the tumor microenvironment [95]. Therefore, the possibility of combining immunotherapy with hormone therapy appears promising. However, relatively few studies have explored this option. Over a decade ago, the monoclonal antibody Tremelimumab (an anti-CTLA4 immune inhibitory molecule) was used in combination with endocrine therapy (exemestane) in metastatic ER-positive patients [96,97]. Unfortunately, the best overall response was not greatly improved after treatment, although around 42% of patients achieved stable disease for more than 12 weeks [97]. More recent studies, however, have shown that new methods can improve outcomes from immunotherapy combined with endocrine therapy [37,49,98]. Some of these studies evaluated the addition of recombinant interferon-beta / interleukin-2 as immunotherapy to first-line salvage hormone therapy (HT) in Erα-positive metastatic breast cancer patients [37,49]. Recombinant interleukin-2 (IL-2) can stimulate lymphokine-activated killer (LAK) cells by stimulating lymphocytes (primarily natural killer cells) [99], and has been used to treat several types of solid and hematological malignancies. In a study assessing responses to hormone treatment in breast cancer patients, those in the control group (hormone (endocrine) treatment: HT) received AIs (letrozole, anastrozole, or exemestane), while the experimental group (hormone-immunotherapy: HIT) also received treatment with interferon-beta/interleukin-2. The PFS in the HIT group was significantly longer than that in the HT group, at 33.1 vs. 18 months. The overall survival (OS) in the HIT group was also more prolonged than in the HT group (median 81 vs. 62 months) [49].

With the development of endocrine therapy, third generation aromatase inhibitors are widely used as the first-line standard drug for postmenopausal women with ER+, HER− advanced or metastatic breast cancer [37,51,100]. Nevertheless, acquired resistance to AIs has been a continuous clinical problem [37]. Since most breast cancers do not benefit significantly from immunotherapy alone, a combination of hormone therapy and immunotherapy was tested [37]. In combination with JD128 (a new type of SERD), immune checkpoint inhibitor (ICI) treatment was found to lead to an enhanced response by increasing immune recognition of tumor cells. This led to increased survival in patients with ER-positive breast cancers. Moreover, even some ER-negative breast cancers including TNBC also responded to treatment, due to the dual role of SERDs in not only regulating hormone levels but also regulating immune cells in the tumor microenvironment [46].

In experimental studies, JD128 significantly inhibited cell transcription and proliferation in MCF-7 cells (human ER+ breast cancer cells) in vitro. It also restrained tumor growth in a xenograft model using MCF-7 cells. Combination therapy with JD128 and anti-PD-L1 antibody significantly inhibited tumor growth in a xenograft model using 4T1 cells, a murine TNBC cell line. Importantly, JD128 and either Fulvestrant or other SERDS with strong antiestrogen activity could stimulate the inhibition of myeloid-derived suppressor cells (ER-positive immune cells) in the tumor microenvironment, and simultaneously promote interactions between ICIs and breast cancer cells by activation of CD8+ and CD4+ T cells. This can result in improved therapeutic outcomes in ER+ breast cancer. 

New endocrine–immunotherapy combination therapies are constantly being studied and developed. In a recent study, two patients with HR-positive metastatic breast cancer benefited from the combination of antiestrogen agents (letrozole or tamoxifen) and immunotherapy (pembrolizumab) [20]. After treatment, the patients showed a higher TCR repertoire, apparently associated with better prognosis after immunotherapy (PFS of more than 21 months). This data appears promising, but the study was small and research is in its early stages, with larger sample sizes required to confirm these results.

While promising, the combination of immunotherapy and hormone therapy is currently in its early stages, and challenges remain to be overcome. For example, in certain cases a decreased rate of survival was reported in patients receiving combination treatment of PD-L1 and hormone therapy. Furthermore, studies using immunotherapy to treat other types of cancer have revealed some potential challenges. For example, in non-small-cell lung cancer patients, the immunosuppressive effects of corticosteroids (such as Prednisone) may reduce the inhibition of PD-L1 or PD-1 (via the use of Pembrolizumab, Nivolumab, Atezolizumab, or Durvalumab) [63,101]. Thus, the combined use of hormone treatment and immunotherapy continues to require further study and optimization.

### 4.4. Cocktail Strategies including Triplet Combinations

Although combination strategies are being developed to improve outcomes of breast cancer and prolong patient survival, challenges remain. Some breast cancer patients suffer from metastases, disease relapse, or therapeutic resistance. One possible reason for treatment failure is the presence of CSCs (anti-cancer stem cells), which can lead to recurrence, metastasis, heterogeneity, and multidrug resistance [102,103]. CSCs have the capacity for self-renewal and differentiation, and can be resistant to some of the double-combination therapeutic strategies described here. Therefore, certain cocktail strategies (triple-combination therapeutic strategies) have been proposed to further optimize outcomes [104].

The combination of Paclitaxel (PTX, a chemotherapeutic agent), Thioridazine (THZ, an anti-CSC agent), and HY1999 (referred to as HY, a PD-1/PD-L1 inhibitor) was shown to have significant anti-cancer efficiency in a mouse metastatic breast cancer model [105]. To optimize the conditions for using these agents, researchers tested three different conditions for treatment of the MCF7 metastatic breast cancer mouse model. The first condition consisted of the three compounds (PTX, THZ, and HY) as free drugs. The second condition involved using nanotechnology. Specifically, a nano device (PM@THL) was employed to load the PTX, THZ, and HY onto a hybrid liposome, resulting in a nanoparticle of 100 nm in size, with a micelle–liposome double-layer structure. PM@THL is sensitive to enzyme and pH values, thus modulating release of the drugs, and targets the tumor by the EPR effect. The third strategy involved loading only two of the drugs (PTX and THZ) onto a liposome, termed TM@TL. The results revealed that this cocktail strategy could improve drug concentrations in the tumors and lungs of mice, and that nanoparticle delivery of the three compounds (PM@THL) was the most effective application method. PM@THL treatment resulted in apoptosis of 92.39% of metastatic cells, and prolonged the survival times of 83.33% of animals to over 60 days, a significant increase compared with the saline group and PM@TL groups. There was a 97.64% rate of suppression of lung metastasis in mice treated with PM@THL. This was the result of the HY blocking the PD-1/PD-L1 connection between T cells and tumor cells, triggering an immune response in which CD4+ and CD8+ T cells infiltrated into the tumor area, eventually killing the bulk of the tumor cells and CSCs. In contrast, in the PM@TL group, tumors regrew 25 days after the last administration, and median survival was 50.5 days. This may have been due to the eventual exhaustion of PTX and THZ. These results indicate that the cocktail strategy is an important approach to the treatment of metastatic breast cancer, although more work is required to optimize the conditions. 

Recently, the inhibition of additional biomarkers has been found to lead to inhibited growth of CSCs, improved outcomes in response to chemotherapeutic treatment, and reduced risk of relapse. These markers include CD44, CD24, ALDH1, and CD133 [106]. CD44 siRNA [107], for example, dramatically improves sensitivity to the chemotherapeutic drug Doxorubicin, by decreasing CD44 expression in CD44+ CD24− breast cancer stem cells. In a clinical trial, treatment with MK-0752 (γ-secretase inhibitor) reduced CSCs in breast tumorgraft models and enhanced the efficacy of the chemotherapeutic drug docetaxel by inhibiting the Notch pathway [108]. Thus, novel combination therapies directed against CSCs may be effective and durable strategies for breast cancer treatment, especially TNBC [109,110].

## 5. Nanotechnology, Computer Technology and New Developing Treatment Options

### 5.1. Nanotechnology

In recent years, with continuous improvements in the field of nanotechnology, nanocarriers have been developed as special vehicles to deliver two or more anti-cancer drugs into tumors by the passive approach of EPR (enhanced permeation and retention) [111], or by an active approach through nanocarriers modified with targeting agents (ligands, aptamers or antibodies) [112]. Drugs delivered by nanocarriers have shown high efficiency, strong immunogenicity, good targeting, and stability when used against several types of cancers [111,113,114]. Results observed in response to nanostructures are often associated with the physicochemical properties of the encapsulated drugs, stability in blood circulation, different patient or tumor types, and other factors [59,103,115].

A preclinical study demonstrated that co-loading lipospheres (nanocarriers) with Cabazitaxel (CBZ) and Thymoquinone (TMQ) could lead to an obvious increase in apoptosis of MDA-MB-231 breast cancer cells, compared with the use of a mixed solution of free CBZ and TMQ. After treatment, dramatic changes in the cancer cells were apparent in the CBZ TMQ lipospheres group, compared with the CBZ TMQ combination solution group, including nuclear fragmentation which eventually led to apoptosis. The above results illustrate the advantages in using nanoparticles for optimal delivery of drugs. Interestingly, nanoparticles may also be beneficial for reversing drug resistance to chemotherapeutic drugs. Zinc oxide nanoparticles, for example, because of their intrinsic toxicity and ability to induce cell apoptosis by autophagy, have been associated with overcoming drug resistance to Doxorubicin (DOX) in MCF-7/ADR cells (multidrug resistant breast cancer cells) [116]. 

### 5.2. Computer Technology

Computational analysis is an important new technology for the discovery of new combination therapies, and for reducing the time involved in testing new strategies. Molecular docking software (such as Autodock Vina) can be used to predict the orientation of one molecule bound to another, which in turn can predict binding affinity. This type of software has become popular because of its powerful functionality, including the ability to analyze the activities of compounds within their pharmacological networks, as well as to explore potential mechanisms of action of various drug combinations [117,118,119,120]. Network pharmacology involves analysis of drugs with respect to the interactome and disease state. This type of technology can be used to help predict compounds that can bind to specific target proteins, as well as for the identification of potential drug targets [121]. Network pharmacology can be integrated with virtual computing, high-throughput omics data analysis, and retrieval of the foundations of network databases, which may predict the combined effects of multiple ingredients, potential targets, and multiple pathways in cancer cells [122]. Computer analysis taking advantage of the availability of enormous somatic genomic profiles and public databases of clinical trials can be used for the rational design of combination therapies. This type of analysis can also provide optimal scheme ranking for new clinical trials, with the possible added advantage of minimizing costs by focusing on the types of treatments that are more likely to be successful [123,124].

In one example involving the use of computer technologies to study drug combinations, Empagliflozin (EMP, an anti-cancer drug) in combination with DOX was found to be an effective novel chemotherapeutic combination for TNBC [118]. EMP is also known as a calmodulin receptor antagonist; it is cytotoxic and can enhance the chemosensitization effect of cells resistant to chemotherapy drugs such as DOX and vincristine [118,125]. The investigators carried out molecular docking studies using the drug design software MOE (Molecular Operating Environment) to evaluate the mechanism by which EMP restrains the binding of calmodulin receptors, by binding important amino acids (Glu84, Glu11, Glu127, and Met144) on calmodulin [118]. Calmodulin is a target protein for anticancer therapeutic intervention, which regulates many of the intracellular actions of calcium, such as cellular proliferation in malignancy [125,126]. Drug synergism studies using CompuSyn software [127] showed the drug reduction index (DRI) values of the DOX/EMP combination to be above 1.0, which indicated a favorable dose reduction in the drug combination compared to monotherapy. The combination index (CI) [127,128] was in the range 0.26–0.15 and the fraction affected values were from 0.5 to 0.95, which indicated synergism in inhibiting the proliferation of MDA-MB-231 TNBC cells by DOX and EMP. Furthermore, cells treated with the DOX/EMP combination displayed remarkable growth inhibition, compared to the control group. Meanwhile, the total percentage of apoptotic cells increased from 27.05 % to 29.22%, compared with DOX alone. The results indicate that a combination of DOX and EMP has a synergistic effect on the inhibition of cell proliferation, migration, and apoptosis. These studies indicate that computer technology can play a crucial role in the identification of potential new combinations of drugs.

## 6. Advantages and Challenges in the Use of Combination Therapy for Breast Cancer Treatment

Surgery, radiotherapy, and chemotherapy remain the primary treatment options for breast cancer. However, new combination therapies are required to treat patients who are unresponsive to current treatments or who develop resistance over time [3,129]. Molecular combination treatments currently under development appear promising for improving outcomes. Like any new treatments, however, they bring advantages along with challenges, summarized briefly below.

### 6.1. Advantages

(a) Precise treatment: Current molecular combination therapy can be directed towards specific targets based on the characteristics of the individual’s tumor [23]. Numerous molecular targeting agents have been explored over the years [115,130], and combination treatment allows the specific targeting of multiple pathways for a precise response. To optimize the therapeutic effects and minimize the likelihood of relapse after surgery, certain combination therapies are currently used as adjuvant or first-line therapy for recurrent or metastatic breast cancer in clinical settings (see Table 2) [130]. (b) Synergistic and additive effects: As discussed throughout this review, molecular combination therapy offers the promise of prolonged survival due to the synergistic or additive effects of using more than one agent [19]. The lack of strong tumor targeting remains a major drawback for monotherapy. Particularly in complicated cancer environments, single drugs may be weak and have an increased chance of leading to off-target effects, compared with combination therapies [84,122]. (c) Reduced side effects and drug resistance compared with chemotherapeutic drugs: Due to the specificity of new and precise combination therapies, side effects can often be minimized. Likewise, some studies showed a reduction in the development of resistance. For example, DOX is an anthracycline antibiotic that is commonly recommended in chemotherapy, but is also associated with cardiotoxicity and drug resistance [131,132,133]. However, when DOX was combined with SR-4835 (CDK12/CDK13 Inhibitor) or Olaparib (PARP Inhibitor), toxicity was reduced and TNBC cell death was enhanced [131].

### 6.2. Challenges

(a) Drug resistance: Although combination therapy can often reduce drug resistance, in some cases patients can become resistant to combination therapies [147,148]. This can involve multiple factors, including a complex tumor microenvironment, drug efflux, cancer stem cells, bulk tumor cells, and crosstalk between signaling pathways. These effects can differ between different subtypes of breast cancer [148]. (b) Drug–drug interaction (DDI) risks and toxicity: With any combination, there is a risk of DDI due to the use of multiple drugs. This can lead to problems such as inflammation, renal or liver failure, thrombocytopenia, brain metastases, denutrition, and sarcopenia. As with any new treatment, the possibilities of side effects and toxicity need to be considered for each combination. A promising possible solution to this problem is the use of molecular docking software to help develop the most favorable combination regimens with the lowest likelihood of DDI [149]. (c) The possibility of low long-term efficiency: Since combination therapies are relatively new, more results from long-term studies are required to assess long-term survival data [150,151]. Combining Bevacizumab or Carboplatin with Anthracycline- and Taxane-based regimens (ATR) increased pCR rates in stage II-III TNBC in a clinical trial [150]. However, one of two updated reports [151] indicated that neither Bevacizumab nor Carboplatin significantly improved long-term outcomes after a median follow-up of 7.9 years. This may be due to the possibility that the immune responses lost power after several years, despite the high complete pathologic response rate during primary treatment. Finally, promising studies that are currently being carried out in animals need to be advanced to clinical trials, to assess their effectiveness in human patients against different subtypes of breast cancer. 

## 7. Conclusions and Future Perspectives

Molecular combination therapy is a promising mechanism for improving outcomes for breast cancer patients, and numerous preclinical and clinical studies have indicated that it can be more effective than single-drug therapy. Combination therapies involve multidisciplinary approaches which may include radiotherapy, chemical therapy, endocrine therapy, targeted therapy, immunotherapy, and other drug therapies. Depending on the stage and subtype of the disease, combination therapy may be the key to achieving individualized treatments with better efficacy than monotherapy, minimizing toxicity, side effects, and recurrence. Nevertheless, challenges remain and it is important to continue to optimize the use and administration of these treatments. In the future, will be important to develop even more comprehensive and personalized treatments for the best possible outcomes. Future use of cancer genomic technology, nanotechnology, and other multidisciplinary data analysis will be critical for the development of the best possible combination treatments.

## Figures and Tables

**Figure 1 ijms-23-11046-f001:**
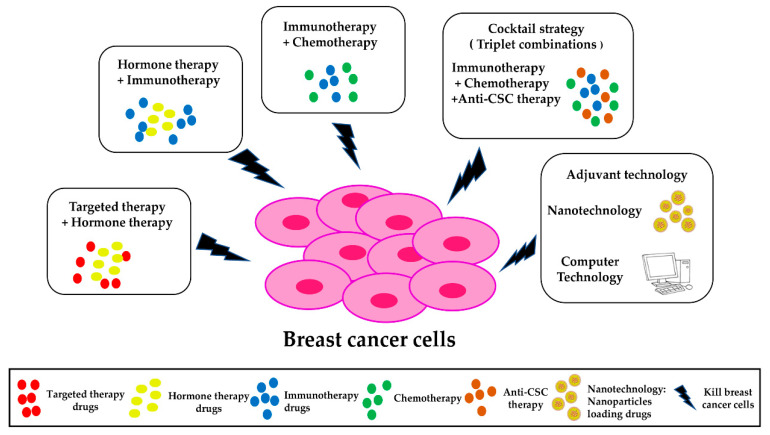
Schematic illustration of current treatments for breast cancer. Traditionally, several different types of treatment have been used, depending on the type and stage of cancer. More recently, combinations of different types of treatments have also been used. Some of the major types of treatments and combination therapies are summarized here.

**Table 1 ijms-23-11046-t001:** Selected drugs commonly used in monotherapies.

**No.**	**Monotherapies**	**Drug Description**	**Mechanism**
	**Targeted Therapy**		
1	Abemaciclib, Ribociclib, Palbociclib	CDK 4/6 Inhibitor	Regulation of Rb phosphorylation, control of cell cycle progression [27].
2	Capivasertib	AKT inhibitor	Inhibition of PI3K/AKT/mTOR signaling, regulation of cell proliferation [1,28,29,30].
3	Everolimus	mTOR inhibitors	
4	Pilaralisib, Alpelisib	PI3K inhibitor	
5	Voxtalisib	dual PI3K/mTOR inhibitor	
6	Bevacizumab	VEGF inhibitor	Anti-VEGF monoclonal antibody, inhibition of blood vessel growth [31,32,33].
7	Olaparib, Talazoparib, Niraparib	PARP inhibitor	Inhibition of PARP, disruption of DNA repair process, increased cancer cell death [27].
**No.**	**Hormone therapy**	**Drug description**	**Mechanism**
1	Letrozole, Anastrozole, Exemestane	AI	Inhibition of aromatase enzyme, leading to inhibition of estrogen production [34].
2	Tamoxifen	SERM	Inhibition of estrogen/ER interaction [35,36].
3	Fulvestrant	SERD	Degradation of ER in breast cancer cells [36,37].
**No.**	**Immunotherapy**	**Drug description**	**Mechanism**
1	Durvalumab, Pembrolizumab, Atezolizumab	PD-1/PD-L1 inhibitor	Inhibition of immune regulatory checkpoints, thus blocking the interaction between T cells and tumor cells [38,39].
2	Pertuzumab, Trastuzumab	Immunotherapy targeted HER2	Inhibition of HER2 signaling pathway, and activation of immune-related responses to HER2 overexpression [40,41,42].
**No.**	**Chemotherapy**	**Drug description**	**Mechanism**
1	Paclitaxel, Docetaxel	Taxanes	Prevention of cell division [43].
2	Doxorubicin, Epirubicin	Anthracyclines	Promotion of DNA damage in cancer cells, leading to apoptosis [44].
3	Cisplatin, Carboplatin	Platinum agents	Interference with DNA synthesis, thus inhibiting cell division [45].

AIs: aromatase inhibitors, mTOR: mammalian target of rapamycin, SERD: selective estrogen receptor down regulator, SERM: selective estrogen receptor modulator, ER: estrogen receptor, CDK 4/6: cyclin dependent kinase 4 and 6, PD-1/PD-L1: programmed cell death-1/programmed death ligand-1, PARP: Poly ADP-Ribose Polymerase, VEGF: vascular endothelial growth factor, HER2: human epidermal growth factor receptor 2, PI3K: phosphatidylinositol-3-kinase.

**Table 2 ijms-23-11046-t002:** Selected current treatments and clinical trials involving the use of combination therapies for breast cancer treatment.

**No.**	**Clinical Trials/Author**	**Trials Phase**	**Status**	**Combination Treatment**				**Patient Population**	**Outcomes (Combination Therapy (CBT) vs.** **Monotherapy)**
				**Targeted** **Therapy**		**Hormone** **Therapy (HT)**			
1	MONARCH 3 (NCT02246621) [100,134,135]	III	First and only CDK4/6 inhibitor approved by FDA in 2021	Abemaciclib (CDK4/6 Inhibitor)		Fulvestrant (SERD) or Anastrozole, Letrozole (AIs)		Postmenopausal HR+, HER2− ABC	PFS: 28.18 months of CBT vs. 14.76 months of HT alone
2	MonarchE (NCT03155997) [135,136]	III	Ongoing	Abemaciclib (CDK4/6 Inhibitor)		Tamoxifen (SERM) or Anastrozole, Letrozole (AIs)		HR+, HER2−, node-positive, high-risk, early BC	2-year IDFS: 92.2% of CBT vs. 88.7% of HT alone, and 25% reduction in risk of recurrence and death
3	BOLERO-4 (NCT01698918) BOLERO-2 (NCT01231659) [50,73,74]	II/III	Completed	Everolimus (mTOR inhibitors)		Letrozole, or Exemestane(AIs)		Postmenopausal women with ER+, HER2− MBC/locally ABC	PFS 22.0 months of CBT vs. 9.0 months of Letrozole alone vs. 3.2 months of Exemestane
4	NCT01082068 [83]	I/II	Completed	Pilaralisib (PI3K inhibitor) or Voxtalisib (dual PI3K/mTOR inhibitor)		Letrozole (AI)		HR+, HER2−, nonsteroidal AI refractory, recurrent, MBC	PFS: 24 weeks for 22% of patients
5	FAKTION (NCT01992952), PALOMA-3 (NCT01942135), MONALEESA-3 (NCT02422615) [28,77,86,87,137,138]	II/III	Ongoing	Capivasertib (AKT inhibitor), or Ribociclib, palbociclib (CDK4/6 Inhibitor), or Alpelisib (PI3K inhibitor)		Fulvestrant (SERD)		Postmenopausal women with HR+, HER2−, or with PI3K/AKT/mTOR mutation, MBC/ABC	PFS and OS of CBT are significantly improved compared with HT alone
**No.**	**Clinical Trials/Author**	**Trials Phase**	**Status**	**Hormone** **therapy**		**Immunotherapy**		**Patient population**	**Outcomes** (**CBT vs.** **monotherapy**)
6	Cases reported [20]		Ongoing	Letrozole or Tamoxifen (Antiestrogen agents)		Pembrolizumab (PD-1/PD-L1 inhibitor)		HR+ MBC	PFS: >21 months
7	PERTAIN (NCT01491737) [67]	II	Ongoing	Anastrozole or Letrozole(AI)		Pertuzumab + Trastuzumab (Immunotherapy targeted HER2)		HER2+, HR+ MBC/locally ABC	PFS: 21.72 months vs. 12.45 months
**No.**	**Clinical Trials/Author**	**Trials Phase**	**Status**	**Targeted** **therapy**		**Immunotherapy**		**Patient population**	**Outcomes (CBT vs.** **monotherapy)**
8	MEDIOLA (NCT02734004) [139,140]	I/II	Ongoing	Olaparib (PARP inhibitor)		Durvalumab (PD-1/PD-L1 inhibitor)		BRCA-mutated MBC	PFS: 8.2 months of CBT vs. 7.0 months of Olaparib alone, vs. 4.2 of Olaparib alone in OlympiAD trial [61].
9	OPACIO (NCT02657889) [141,142]	I/II	Completed	Niraparib (PARP inhibitor)		Pembrolizumab (PD-1/PD-L1 inhibitor)		Advanced or metastatic TNBC, ovarian cancer	Median PFS: 8.3 months, with a objective response rate of 21%, and disease control rate of 49%
**No.**	**Clinical Trials/Author**	**Trials Phase**	**Status**	**Immunotherapy**		**Chemotherapy** (**CT**)		**Patient population**	**Outcomes** (**CBT vs.** **monotherapy**)
10	KEYNOTE-522 (NCT03036488)KEYNOTE-355 (NCT02819518) [92]	III	First-line treatment approved by FDA in 2021	Pembrolizumab		nab-paclitaxel, paclitaxel, or gemcitabine plus carboplatin		TNBC	PFS: 9.7 months of CBT vs. 5.6 months of CT alone
11	IMpassion130 (NCT02425891) [62]	III	First-line treatment approved by FDA in 2021	Atezolizumab		Albumin-bound paclitaxel (nab-paclitaxel)		Advanced TNBC	PFS: 9.3 months of CBT vs. 6.1 months of CT alone; OS: 28.9 months of CBT vs. 20.8 months of CT alone.
12	IMpassion131 (NCT03125902) [143]	III	Not approved by FDA in 2020	Atezolizumab		Paclitaxel		Advanced/metastatic TNBC	PFS or OS are not improved vs. CT alone, with potential safety concerns.
**No.**	**Clinical Trials/** **Author**	**Trials Phase**	**Status**	**Targeted** **therapy**		**Chemotherapy**		**Patient population**	**Outcomes** (**CBT vs.** **monotherapy**)
13	NCT02456857 [144]	II	Ongoing	Bevacizumab (VEGF inhibitor) + Everolimus (mTOR inhibitors)		Doxorubicin		Locally advanced TNBC with insensitivity to standard chemotherapy	The objective response rate was 21%
14	NCT01281696 [145]	II	Completed	Bevacizumab (VEGF inhibitor)		Etoposide, Cisplatin		BC with brain metastases	PFS: 9.1 months, OS: 10.7 months
15	OlympiAD (NCT02000622) [61]	III	Ongoing	Olaparib (PARP inhibitor)		Capecitabine, Eribulin, orVinorelbine		Metastatic breast cancer with BRCA mutation	PFS: 7.0 months vs. 4.2 months of chemotherapy alone
**No.**	**Clinical Trials/** **Author**	**Trials Phase**	**Status**	**Cocktail Strategy** (**Triplet combinations**)				**Patient population**	**Outcomes** (**CBT vs.** **monotherapy**)
				**Immunotherapy**	**Chemotherapy**		**Anti-CSC** **therapy**		
16	Lang, et al. [105] Preclinical study			HY19991 (HY, PD-1/PD-L1 inhibitor)	Paclitaxel (PTX)		Thioridazine (THZ)	MCF-7 MBC mice treated with PTX/THZ/HY liposome (PM@THL)	The tumor inhibiting rate of PM@THL was 93.45%
				**Immunotherapy**	**Chemotherapy**		**Target therapy**		
17	HER2CLIMB (NCT02614794) [40,146]	III	First-line treatment approved by FDA in 2020	Trastuzumab	Capecitabine		Tucatinib (tyrosine kinase inhibitor)	Advanced or HER2+ ABC or MBC	Better PFS, OS, and safe

BC: breast cancer, MBC: metastatic breast cancer, ABC: advanced breast cancer, HR: hormone receptor (estrogen and progesterone receptors), TNBC: triple-negative breast cancer, HR+: hormone receptor positive, HER2+: HER2-positive, HER2−: HER2-negative, PFS: progression-free survival, OS: overall survival, AIs: aromatase inhibitors, mTOR: mammalian target of rapamycin, HT: hormone therapy, CBT: combination therapy, CT: Chemotherapy, IDFS: invasive disease-free survival, SERD: selective estrogen receptor downregulator, SERM: selective estrogen receptor modulator.

## Data Availability

This is a review article, and no new unpublished data is included.

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
