# Peer review of "Current Molecular Combination Therapies Used for the Treatment of Breast Cancer"

_ijms, 2022, doi:10.3390/ijms231911046_

Round 1

Reviewer 1 Report

Congratulations on a comprehensive review of the various combination therapies used to treat breast cancer. Overall, the manuscript is well-structured and well-written. It discusses the major ongoing combination therapies, such as molecular targeted therapy, hormone therapy, immunotherapy, and others, in different breast cancer subtypes, as well as the associated challenges. I enjoyed reading this review and would recommend it for publication.

Some minor suggestions:

1.       A thorough visual representation (pictorial) would make it much simpler to understand the perspective and broaden its appeal.

2.       Page 1, Line 30: I think the author intended to write “Luminal B”.

3.       Although the authors mentioned anti-CSC therapy, I would suggest a section describing recent developments in the use of specific stem-cell markers, like CD44, GD2, ALDH, etc., to target cancer stem cells, which are being investigated in conjunction with traditional therapies.

Author Response

Please see the response to reviewer 1's comments in the attachment. Thank you!

Reviewer 2 Report

Generally, the review has a clear and logical narrative, and is written in a concise and exact fashion that provides an excellent introduction to the field of "difficult therapies".  There are no major deviations or dead ends or unnecessary sections anywhere in the text. It is also logical and recommended that first monotherapies are introduced and discussed, before these treatments/drugs are then recapitulated in the section on combination therapies (the central topic of the work). Nevertheless, prior to table 1, another table could be added summarizing the most widely used drugs, and their mechanisms of action/pathway, before these points are then discussed in the combination therapy table. Generally, there may be a strong interest in some of the readers to know about which are the central, molecular targets of these drugs that are used together with others, in particular, as it has been stated that no combinations are being discussed that cover the same pathways. Then, tables like table 1 that pass over 3 pages tend to be - sorry for saying this - rather boring and are not really read by most readers, anyway. There could be other forms of presenting these data, maybe? Instead of focusing on the NCT numbers, maybe the table could be organized by drugs, or by pathways covered. This is partly done in form of the subdivisions of the table, but not consistent and may be less clear than it could be. That way, table 1 would also be more in line with the clear structure of chapters 3 and 4. I also wonder, how many drugs (for each of the targets/pathways covered) are actually approved by FDA or EMA etc? That may be interesting, to see the scope and the many options of therapies available in breast cancer (quite in contrast to other diseases, such as head and neck cancers, where there are much less options available). Overall, this plethora of options is rather impressive in BrCa, but the reader cant really appreciate it, if this is not highlighted properly. 

The literature review is definitely up to date, with most of the publications/references cited no less than 5 years old, and the rest less than 10 years old. This apparently represents a very reasonable and comprehensive overview of the newer developments in the field, with less focus on the "historic", earlier progress made before. Which is optimal, in my opinion, for this kind of review. A total of 122 references are cited in the text, which is not a huge amount but speaks for effective prioritization and good literature research work. 

There are very few redundant sections or sentences, but some do exist and could be shortened ... for example here (lines 80 - 83):  "We do not discuss, for example, combination therapies in which two or more chemotherapy drugs are used, or in which chemotherapy drugs are combined with radiation [19]. Instead, we focus on combination therapies in which different intracellular pathways are targeted and by different mechanisms."

The introduction (chapter  1) is a brief but comprehensive overview of the subtypes of breast cancers, which quickly focuses on the more difficult-to-treat subtypes and which clinical options may exist to improve therapy outcomes. 

Chapter 4 really is the core of the entire review, and is rather convincing as a summary of "options available" for patients. There is little to criticize this central part, in my opinion; maybe again the possibility to align this listing of combinations with table 1 in a somewhat more obvious, intuitive fashion. Since the majority of combination therapies still contain at least 1 endocrine treatment, this may be a good organizing line. 

Chapter 4.4 is (in my opinion) not fitting here at all, as it's entirely experimental; no such drugs have been approved, and if it should be retained at all, then as a separate chapter pointing to "experimental combination therapies" or something like that. Otherwise, it's not a good fit for the narrative of the entire manuscript. 

The same applies to chapter 4,6; both 4.4 and 4.6 are set apart from the rest of the paper any should be combined (maybe) as "future aspects" or "developmental issues"... 

In contrast, I think that chapter 4.7 (pros and cons) could be expanded and highlighted more. There is a clear need to cover these risks, and probably not enough research, but the issues should be raised and there is no better place for doing this than a review like this one. 

Author Response

Please see the response to reviewer 2's comments in the attachment. Thank you!

Reviewer 3 Report

This review paper by Wang and Minden gives a comprehensive description of clinical trials using combination therapies in breast cancer. They focus their description on combination therapies drugs targeting different pathways. As stated, they don’t discuss, combinations that involve two or more chemotherapy or combination with radiotherapy. Authors propose the use of nanotechnology and computer technology to standardize and optimize combination of drugs and their potential effects on cancer patient. These last two sections suffer from some weaknesses as they were not well described and lack a deep review of the recent literature with convincing examples form published study or trials. Authors might edit these two sections and illustrate with figures to give a clear understanding of how nanotechnology and computer technology can help improving drug combinations in breast cancer

Author should correct several mistakes and clarify some sections.

Comments:

1)    Please specify or explain more how Normal-like breast cancer can be distinguished from lumina A breast cancer in line: 33 “Normal-like 33 breast cancer usually shares characteristics with Luminal A breast cancer but has some 34 differences in overall genetic makeup [8-10].  

2)    Line 133; “Such therapies include PI3K/AKT/mTOR, AMPK inhibitors,” why authors are citing AMPK inhibitors for breast cancer here?  If it is real, which inhibitor of AMPK is used in breast cancer?  

3)    In the section 3.3 immunotherapies authors can describe more trastuzumab-DM1 and cite some references as described for trastuzumab.

4)    Line 190, “In a phase 2 clinical trial, the PFS increased from 22.0 months to 9.0 months when the combination treatment was used, compared with Letro-191 zole alone [30,47,48]. Please check “increased from 22.0 months to 9.0 months” as it is quite confusing.

5)    Line199:” Combining endocrine therapy (such as Etrozole, Anastrazole, or Fulvestrant), with CDK 4/6 inhibitors (such as Palbociclib or Abemaciclib), was beneficial for decreasing endocrine sensitivity in patients with HR+, HER2-, advanced, or metastatic breast cancer [50]”

Please check “..was beneficial for decreasing endocrine sensitivity”, the meaning is not clear, please check whether it’s increasing sensitivity instead of decreasing sensitivity.

6)    Line 202: “The median PFS for patients in the combination therapy group was 28 months, compared to 14.9 months for the placebo plus endocrine therapy group. The median PFS in combination therapy group was 22.4 months compared to endocrine therapy alone, and this is likely to be higher, due to patients who were still surviving at the time the report was made [50].”  These two sentences need to be re-edited as it seems confusing and contradictory. What is the difference between PFS of 28 months in the first sentence and 22.4 months in the second? What is the difference of PFS in placebo plus endocrine therapy group in the first phrase and in endocrine therapy alone in the second phrase?

7)    In line 208, please correct: Alpelisib is not a CDK4/6 inhibitor.

8)    Line 213: “For example, in a separate study, adding Palbociclib was added to endocrine therapy (either anti-estrogen or AI)..” please delete “was added”

9)    Line 221: “The PI3K inhibitor (Pilaralisib) or a dual PI3K/ mTOR inhibitor (Voxtalisib) in combination with Letrozole (an AI) was shown to be effective in patients who have HR+, HER2-, nonsteroidal AI refractory, recurrent or metastatic breast cancer in clinical trials [57]. The median PFS of the combination group was 7.9 weeks, and 22% of patients had a PFS of 24 weeks.” Please clarify the second phrase, which one of the combinations (Pilaralisib + AI) or Voxtalisib + AI) gives a PFS of 24 weeks?

10) Line 318: “In animal studies, using in vivo MCF-7 (HR+ breast cancer cells) xenograft models, combination therapy with JD128 and anti-PD-L1 antibody significantly inhibited tumor growth. Importantly, JD128, and either Fulvestrant or other SERDS with strong antiestrogens activity could stimulate the inhibition of myeloid-derived suppressor cells (ER-positive immune cells) in the tumor microenvironment, and simultaneously promote the interactions between ICIs and breast cancer cells by activation of CD8+ and CD4+ T cells. This results in improving therapeutic outcomes in ER+ breast cancer.”  Please add a reference to this paraph. If it’s the reference number 24, please explain how using an in vivo model of MCF7 cells that are human cell line that only grow in immunodeficient mice could be used in an immunotherapy study?  

11) Lien 339: “ Thus, the combined use of hormone treatment and immunotherapy therapy still requires further study and optimization” please delete “therapy after immunotherapy”

12) Line 351: please replace “A new trial” with “Preclinical study”  

13) Line 159, please correct LD-L1 to PD-L1

Author Response

(The authors gave the same response as above.)

Round 2

Reviewer 3 Report

I'm satisfied with the corrections and approve this review paper